# Interaction of Styrylpyridinium Compound with Pathogenic *Candida albicans* Yeasts and Human Embryonic Kidney HEK-293 Cells

**DOI:** 10.3390/microorganisms9010048

**Published:** 2020-12-27

**Authors:** Simona Vaitkienė, Laura Bekere, Gunars Duburs, Rimantas Daugelavičius

**Affiliations:** 1Department of Biochemistry, Faculty of Natural Sciences, Vytautas Magnus University, Kaunas LT-44404, Lithuania; rimantas.daugelavicius@vdu.lt; 2Latvian Institute of Organic Synthesis, Riga LV-1006, Latvia; laura.bekere.8@gmail.com (L.B.); gduburs@osi.lv (G.D.)

**Keywords:** styrylpyridinium, *Candida albicans*, HEK-293, adhesion, cytotoxicity, gene expression, multidrug resistance, efflux pumps

## Abstract

*Candida albicans*-caused local and systemic diseases are a serious health issue worldwide, leading to high mycosis-associated morbidity and mortality. Efficient combinations of novel compounds with commonly used antifungals could be an important tool for fighting infections. The aim of this study was to evaluate the interaction of synthesized 4-(4-cyanostyryl)-1-dodecylpyridin-1-ium (CSDP^+^) bromide alone or in combination with fluconazole with yeast and mammalian cells. We investigated cytotoxicity of the tested agents to mammalian HEK-293 cells and the influence of CSDP^+^ on the ability of *C. albicans* wt and a clinical isolate to adhere to HEK-293. Accumulation of lipophilic cation ethidium (Et^+^) was used to monitor the activity of efflux pumps in HEK-293 cells. The effect of CSDP^+^ on the expression of the main efflux transporter genes and transcription factors in *C.*
*albicans* cells as well as HEK-293 efflux pump gene *ABCB1* was determined. The study showed that CSDP^+^ alone and in combination with fluconazole was nontoxic to HEK-293 cells and was able to reduce *C.*
*albicans* adhesion. The treatment of *C.*
*albicans* cells with CSDP^+^ in combination with fluconazole resulted in a considerable overexpression of the *MDR1* and *MRR1* genes. The findings suggest that these genes could be associated with efflux-related resistance to fluconazole. Measurements of Et^+^ fluorescence and analysis of *ABCB1* gene expression demonstrated that mammalian cells were not sensitive to concentrations of CSDP^+^ affecting *C. albicans*.

## 1. Introduction

Resistance to antifungal treatment, especially to the drugs of the azole class, is a serious problem in the case of fungal pathogen *Candida albicans*. Various molecular mechanisms of resistance in susceptible and resistant clinical isolates have been discussed [1,2]. An important factor involved in *C. albicans* pathogenesis is the ability of yeasts to adhere to host cells and to change the cell shape from ellipsoidal to filamentous [3,4,5]. *C. albicans* adhesion to host tissues contributes to biofilm formation and increases virulence. However, relatively little is known about molecular mechanisms leading to *C. albicans* infections in human tissues and adhesion to medicinal materials.

In our previous study, we have demonstrated [6] that styrylpyridinium compounds, for a few decades known as fluorescent probes [7,8] or for their antimicrobial properties against bacteria [9], efficiently inhibited the growth of *C. albicans* cells. The most active styrylpyridinium compound 4-(4-cyanostyryl)-1-dodecylpyridin-1-ium (CSDP^+^) bromide inhibited the growth and respiration of *C. albicans* cells very effectively. In addition, a strong synergistic effect was demonstrated when CSDP^+^ was used in combination with fluconazole. At the same time, the results of our experiments showed that in the absence of functioning efflux pump genes, *C. albicans* cells were more susceptible to CSDP^+^ [6]. These results are consistent with the data [10] that *C. albicans* resistance to antifungal drugs is associated with the reduced intracellular concentration of these compounds.

It is well known that the genes of ABC transporters *CDR1* and *CDR2*, encoding ATP-dependent efflux pumps, are overexpressed in resistant *C. albicans* strains, especially in those not sensitive to the drugs of the azole class. Deletion of these genes causes hypersensitivity to azoles [1,11,12]. Expression of ABC transporters Cdr1 and Cdr2 is controlled by the zinc cluster transcription factor TAC1, directly binding to drug-responsive elements in gene promoters. In general, “gain-of-function” mutations induce hyperactivity of the transcription factor TAC1, increasing the expression of *CDR1* and *CDR2* [13,14].

The *MDR1* gene encodes a transporter of the major facilitator superfamily that uses the proton motive force for the extrusion of drugs and other noxious compounds from cells. Overexpression of this pump causes resistance, and deletion of the transporter gene results in the hypersensitivity of cells to azoles [1]. MRR1, a zinc cluster transcription factor, mediates the expression of *MDR1* in response to different chemicals, and overexpression also causes the activation of “gain-of-function” mutations in the MRR1 transcription factor [15].

Resistance of mammalian tissues to lipophilic and amphiphilic compounds also involves ABC transporters. The best characterized MDR genes in mammalians include *ABCB1* (also known as *MDR1* or *P-glycoprotein*), *ABCC1* (also known as *MRP1*), and *ABCG2* (also known as *BCRP* or *MXR*) [16]. However, we need more information how mammalian cells interact with antifungals.

Due to the lack of information about the mechanisms of action of styrylpyridinium compounds, we designed this study to evaluate the effects of CSDP^+^ on the ability of *C. albicans* to adhere to human kidney embryonic cells HEK-293 and to affect the expression levels of efflux pump genes in *C. albicans* and HEK-293 cells. The study revealed that CSDP^+^ alone and in combination with fluconazole reduced *C. albicans* adhesion and resulted in a considerable overexpression of the *MDR1* and *MRR1* genes.

## 2. Materials and Methods 

### 2.1. Chemistry

Synthesis of CSDP^+^ from 4-methyl-1-dodecyl-4-methylpyridinium-1 bromide and 4-cyanobenzaldehyde was described in our previous study [6]. 

### 2.2. Strains and Cell Lines

*C. albicans* strain ATCC10231 (wild type, wt) and clinical isolate SV1 were studied. Yeast cultures were grown in YPD media (1% yeast extract, 2% peptone, 2% glucose) at 37 °C for 18 h.

Human embryonic kidney cell line HEK-293 was used in this study. HEK-293 cells were grown in minimal essential amino acids medium (MEM), supplemented with fetal bovine serum (10% *v*/*v*; GIBCO; Life Technologies, Grovemont Cir, Gaithersburg, MD, USA), streptomycin/penicillin (2% *v*/*v*; GIBCO; Life Technologies, Grovemont Cir, Gaithersburg, MD, USA), and Amphotericin B (1% *v*/*v*; GIBCO; Life Technologies, Grovemont Cir, Gaithersburg, MD, USA) in 75-cm^2^ Falcon culture flasks under standard conditions (5% CO_2_ in air at 37 °C).

### 2.3. Adhesion Assay

The adhesion assay was carried out as described elsewhere [17], with some modifications. HEK-293 cells were grown in 75-cm^2^ Falcon culture flasks in MEM, and after reaching confluence, the adhesion of *C. albicans* was assessed. The amount of 1 × 10^5^ cells/well was used to inoculate MEM-containing 24-well tissue culture plates and incubated at 37 °C for 24 h to reach a confluent monolayer. Before the experiments, yeast cells were grown in yeast extract-peptone-dextrose (YPD) medium for 18 h, pelleted at 3000× *g* for 10 min, and resuspended in MEM medium without antimicrobials. HEK-293 cells, cultured for 24 h, were washed three times with phosphate-buffered solution (PBS, pH 7.5). Yeast suspension (100 μL, 1 × 10^6^ cells) was inoculated into plate wells containing antimicrobials. CSDP^+^ (0.25–16 μg/mL), fluconazole (0.0625–0.5 μg/mL), or combination of 0.25-μg/mL CSDP^+^ and 0.0625-μg/mL fluconazole were used for the adhesion assay.

*C. albicans* cells were maintained in contact with HEK-293 for 2 h. After this procedure, the wells were washed three times with PBS to remove unattached yeasts, and HEK-293 cells were detached using 1-mL trypsin-EDTA (Gibco) diluted in PBS, incubating at 37 °C for 5 min. Aliquots of 20 μL were taken, serially diluted, inoculated onto SDA plates, and incubated for 24 h. Finally, the number of yeasts adhered to HEK-293 cells was assessed by counting the colony-forming units (CFU). In the control group, yeast suspensions were not treated with any compounds. 

### 2.4. Evaluation of Cytotoxicity 

To evaluate cytotoxicity, HEK-293 cells were grown in MEM as described in the adhesion assay. After reaching sufficient confluence, the cells were passaged by trypsinization. Effects of the studied compounds on HEK-293 cells were examined using a 2,3-bis [2-methoxy-4-nitro-5-sulfophenyl]-2H-tetrazolium-5-carboxanilide (XTT) viability assay. This assay is based on the ability of metabolically active (live) cells to reduce XTT tetrazolium salt, leading to the formation of orange-colored formazan compounds. 

HEK-293 cells were grown in 96-well plates, starting from 1 × 10^4^ cells per well. After 24-h incubation, the culture medium was carefully aspirated (avoiding damage of the attached cells), and wells were gently washed with PBS two times. Solutions of CSDP^+^, fluconazole, or combination of both agents were added to the plate wells and incubated at 37 °C for 24 h. After incubation, the medium was removed from wells, and 100 µL of fresh MEM and 50 µL of XTT solution were added to each well. The plates were covered to avoid direct light and incubated at 37 °C for 2 h to induce formation of the formazan dye. After incubation, the plates were gently shaken, and absorbance was spectrophotometrically registered at 492 nm using a TECAN GENios Pro™ plate reader. 

Viability was calculated according to the equation:


$\mathrm{Viability}\;\mathrm{of}\;\mathrm{cells}\;\left(\%\right)=\frac{\mathrm{absorbance}\;\mathrm{of}\;\mathrm{cells}\;\mathrm{treated}\;\mathrm{with}\;\mathrm{compounds}}{\mathrm{absorbance}\;\mathrm{of}\;\mathrm{untreated}\;\mathrm{cells}}\times 100\;\%$


### 2.5. Fluorescence Measurement

Accumulation of ethidium (Et^+^), well known as an efflux pump substrate, was used to monitor the efflux activity in HEK-293 cells. Stock solution of ethidium bromide (EtBr; Sigma Aldrich, USA) was added to cell suspensions to the final concentration of 0.5 µM. The samples were mixed, and volumes of 120 μL were transferred into 96-well flat-bottom black plates (60,000 cells per well) within 2 min. At the same time, CSDP^+^ and fluconazole, as well as their combination, were added to the wells. Fluorescence was monitored by using a TECAN GENios Pro™ (Männedorf, Switzerland) reader at an excitation wavelength of 535 nm and emission wavelength of 612 nm, thermostating the plate at 37 °C. After 30 min, detergent digitonin was added to all wells to the final concentration of 0.1 mg/mL. Measurements were continued for 10 min to obtain the maximum binding of Et^+^ to DNA, leading to the maximal Et^+^ fluorescence. In controls, the cells were not treated with any compound, or digitonin was added to the cell suspensions before EtBr.


$\mathrm{Accumulation}\;\mathrm{of}\;\mathrm{Et}+\left(\%\right)=\frac{\mathrm{fluorescence}\;\mathrm{of}\;\mathrm{measurement}\;\mathrm{point}\;\mathrm{every}\;5\;\min}{\mathrm{fluorescence}\;\mathrm{at}\;\mathrm{the}\;\mathrm{last}\;\mathrm{point}\;\mathrm{of}\;\mathrm{the}\;\mathrm{measurement}\;}\times 100\;\%$


### 2.6. Analysis of Gene Expression

The expression of *C. albicans* efflux pump *CDR1*, *CDR2*, and *MDR1*, and transcription factor *MRR1* and *TAC1* genes as well as the expression of HEK-293 efflux pump gene *ABCB1* were evaluated after exposure to CSDP^+^, fluconazole, and combination of both compounds. 

Yeast cultures were grown for 18 h in the YPD medium under different conditions: in the presence of the tested compounds or in the absence of any compounds. Pellets of overnight cultures were collected by centrifugation at 4 °C for 10 min at 3000× *g* (Heraeus Megafuge 16R, ThermoFisher Scientific, Osterode am Harz, Germany). HEK-293 cells, starting with 60,000 cells/well, were grown in 6-well plates under the same conditions as yeast cells: in the presence of CSDP^+^, fluconazole, and their combination or in the absence of any compounds. 

Total RNA from SV1 and wt cells was extracted using a Trizol Plus RNA kit (Invitrogen, Carlsbad, CA, USA) with ZR BashingBead Lysis Tubes (Zymo Research, Irvine, CA, USA). For HEK-293 RNA isolation, the same kit, but without glass beads, was used following the instructions. cDNA synthesis was performed using a High Capacity cDNA Reverse Transcription Kit (Applied Biosystems, Carlsbad, CA, USA), according to the manufacturer’s instructions. RT-PCR was performed using StepOne^TM^ and StepOnePlus^TM^ Systems (Applied Biosystems, Foster city, CA, USA) in 20-µL reaction volume containing the Power SYBR Green RT-PCR Mix (ThermoFisher Scientific, Osterode am Harz, Germany), cDNA template, forward and reverse primers (Table 1, ThermoFisher Scientific, Osterode am Harz, Germany). RT-PCR conditions for *CDR1*, *CDR2*, *MDR1*, *MRR1*, and *TAC1* genes were as follows: 95 °C for 5 min, followed by 40 cycles at 95 °C for 15 s, and 60 °C for 30 s. RT-PCR conditions for *ABCB1* were as follows: 95 °C for 5 min, followed by 40 cycles at 95 °C for 15 s, and 58 °C for 30 s. To determine the expression of genes of interest, the differences (∆) between threshold cycles (Ct) were measured. Data were presented as a fold change in gene expression normalized to the *18S* gene (for yeast) and *GAPDH* gene (for HEK-293) as a control.

### 2.7. Statistical Analysis

Statistical analysis was performed using the GraphPad Prism 8 software (ver. 8.01, 2018; GraphPad Software Inc., San Diego, CA, USA). All the experiments were performed in three biological replicates with three technical repetitions. Results are expressed as means ± standard deviation (SD).

## 3. Results and Discussion

Our previous studies demonstrated that CSDP^+^ acted fungicidally against *C. albicans* wt, *cdr1*Δ, *cdr2*Δ, and *cdr1*Δ*cdr2*Δ strains, and the higher concentration was required to inhibit the growth of SV1 strain cells. MIC_50_ of CSDP^+^ for wt cells was 2.8 μg/mL, and MIC_90_ was 8 μg/mL. The SV1 strain demonstrated increased resistance to this compound and 16 μg/mL was required to inhibit the growth of these cells 90%. MIC_50_ and MIC_90_ of fluconazole for both yeast strains were 0.15 μg/mL and 0.5 μg/mL, respectively. Combination of these two compounds showed a relevant synergism with ΣFIC of 0.19 for SV1 strain cells and 0.38 for wt ones. Treatment with MIC_90_ of CSDP^+^, needed to inhibit the growth of *C. albicans* SV1 cells by 90%, resulted in a 30% decrease in the viability of CHO-K1 cells, but fungicidal concentration (1 μg/mL) of fluconazole in combination with 0.25 μg/mL of CSDP^+^ reduced viability of CHO-K1 cells only by 10%. The results suggested that CSDP^+^ could be a promising agent for combined antifungal treatment [6]. 

### 3.1. Adhesion of C. albicans to HEK-293 Cells

The binding of *C. albicans* to the surface of host cells is an important step in the process of infection, and its ability to adhere is a critical feature in the processes of colonization and persistence in tissues [17,18]. Therefore, one of the promising approaches to reduce yeast-caused infections is the development of adhesion-preventing agents. We aimed to find out how CSDP^+^, fluconazole, and their combination impact the adhesion of *C. albicans* wt and SV1 strain yeasts to human embryonic kidney cell line HEK-293 cells.

The results showed that in the presence of 0.25 µg/mL of CSDP^+^, the adhesion of *C. albicans* to the surface of HEK-293 cells decreased by 34% and 32% for wt and SV1 strain cells, respectively (Figure 1). After the increase of CSDP^+^ concentration to 16 µg/mL, the adherence of wt and SV1 strain yeasts to HEK-293 cells decreased by 54% and 36%, respectively. The effect of fluconazole on the adhesion of *C. albicans* to HEK-293 cells was negligible. However, a 60% and 40% decrease in the adherence of *C. albicans* wt and SV1 strain cells, respectively, was documented when the combination of low concentrations of fluconazole and CSDP^+^ was applied (Figure 1). 

Summarizing our results, we can conclude that CSDP^+^ alone and in combination with fluconazole markedly reduced the ability of *C. albicans* to adhere to HEK-293 cells. 

### 3.2. Cytotoxicity to HEK-293 Cells

In the following experiments, we verified the effects of CSDP^+^ and fluconazole on the viability of HEK-293 cells. Viability of HEK-293 cells gradually diminished to 74% when the concentration of CSDP^+^ increased from 0.25 to 16 µg/mL. However, at the concentration of 64 µg/mL, this compound reduced the viability of HEK-293 cells to 15% (Figure 2). In the presence of fluconazole (up to 1 µg/mL) or combination of these compounds, the survival rate of HEK-293 cells ranged between 90% and 100% (Figure 2). 

Summarizing, CSDP^+^ at concentrations up to 16 µg/mL as well as fluconazole or combination of these two drugs showed no significant effect on the viability of HEK-293 cells. The results of these experiments support our previous findings with hamster cells [6] demonstrating that CSDP^+^ has the weakest effect on the viability of CHO-K1 cells among various styrylpyridinium compounds. 

### 3.3. Effect of CSDP^+^ and Fluconazole on Et^+^ Fluorescence in HEK-293 Cell Suspensions

A fluorescent probe EtBr is widely used as a substrate for studies on efflux pump activities in bacterial and eukaryotic cells [19,20]. In aqueous solutions, the fluorescence of Et^+^ is very weak, but it considerably increases when this indicator gets into cells, mainly due to the intercalation into DNA or dsRNA [21]. Efflux pumps remove Et^+^ from the cells, and the intensity of fluorescence correlates well with the activity of extrusion [22,23]. Here we investigated the effects of CSDP^+^ and fluconazole on the efflux of Et^+^ from HEK-293 cells. 

At a concentration of 0.25 µg/mL, CSDP^+^ alone or in the presence of fluconazole, the intensity of Et^+^ fluorescence was close to the level of control cells, incubated without any additions (Figure 3). After the exposure of HEK-293 cells to 16 µg/mL of CSDP^+^, the intensity of fluorescence was 10%–20% higher, and at the concentrations of 64–128 µg/mL, CSDP^+^ was able to increase Et^+^ fluorescence close to the level of control cells in the presence of non-ionic detergent digitonin (Figure 3). Digitonin is a detergent permeabilizing the plasma membrane. The maximum fluorescence of Et^+^ was achieved after digitonin addition, indicating the maximum binding of this lipophilic ion to DNA [24].

The obtained results suggest that CSDP^+^ at low concentrations (<16 µM) does not damage HEK-293 cells, and efflux pumps efficiently prevent Et^+^ entry. However, at the concentrations of CSDP^+^ of more than 64 µg/mL, Et^+^ gets easier into the cells because of the increased permeability of the plasma membrane or the inhibition of efflux. 

### 3.4. Expression of efflux pump genes in C. albicans 

Our previous study [6] demonstrated that the expression of *CDR1*, *CDR2*, and especially *MDR1* genes in SV1 cells was significantly higher as compared with wt cells. It is known that the overexpression of *MDR1* is a factor promoting the resistance of pathogenic *C. albicans* to fluconazole and other toxic compounds [15]. In clinical *C. albicans* isolates, *MDR1* overexpression is usually accompanied by the upregulation of other genes. For example, *C. albicans* strains that had become fluconazole-resistant due to constitutive *MDR1* upregulation contained also mutations in *MRR1* gene, which controls *MDR1* expression [25]. 

The results of our experiments indicated that wt cells treated with 2 µg/mL of CSDP^+^ demonstrated a mild increase in the expression of *CDR1* and *CDR2* genes, while no significant changes in the expression of *TAC1* were documented compared with untreated cells (Figure 4A). The slight activation of the *MDR1* gene was also observed, but the greatest increase (44.36 ± 3.12-fold) was documented for *MRR1* expression (Figure 4A). 

Cell treatment with 0.125 µg/mL of fluconazole caused a significant increase in the expression of *CDR1*, *CDR2*, and *TAC1* genes as compared with control cells (Figure 4A). As in the case of pretreatment with CSDP^+^, an elevated expression of *MDR1* and overexpression of *MRR1* were observed in fluconazole-treated wt cells (Figure 4A). Exposure of wt cells to 0.0625 µg/mL of fluconazole and 0.25 µg/mL of CSDP^+^ also slightly increased the expression of *CDR1* and *CDR2* (Figure 4A). Combination of these compounds had no significant effect on *TAC1* expression (Figure 4), but markedly increased *MDR1* and *MRR1* expression (31.65 ± 2.64 and 60.22 ± 3.93-fold, respectively; Figure 4A). 

Further, we evaluated the effect of CSDP^+^ and fluconazole on the expression of *CDR1*, *CDR2*, *MDR1*, *TAC1*, and *MRR1* genes in *C. albicans* cells of SV1 strain. In this case after cell treatment with 2 µg/mL of CSDP^+^, there were no significant changes in the expression of *CDR1* and *TAC1* genes (Fig. 4B). Meanwhile, the expression of *CDR2* significantly increased after CSDP^+^ treatment (Figure 4B). CSDP^+^ also caused an elevation of *MDR1* and *MRR1* expression (Figure 4B). The same tendency was observed in the case of fluconazole: expression levels of *CDR2* as well as *MDR1* and *MRR1* were increased compared with control (Figure 4B). 

SV1 cells treated with the combination of both agents showed a 2.0 ± 0.28-fold and a 6.08 ± 0.48-fold increase in the expression of *CDR1* and *CDR2* genes, respectively, as compared with untreated cells (Figure 4B). A mild increase in *MDR1* expression was also observed (Figure 4B). Similarly, as in the case of wt cells, the greatest overexpression, up to 20-fold, was observed for the *MRR1* gene after the exposure of SV1 cells to the combination of CSDP^+^ and fluconazole (Figure 4B). 

Under normal conditions, the *MDR1* gene is usually not expressed in fluconazole-susceptible isolates [26]. However, it was demonstrated that *C. albicans* isolates from abdominal fluid, susceptible to fluconazole but mildly resistant to itraconazole, had a high expression of the *MDR1* gene [1,27]. Moreover, it was recorded that the deletion of *MRR1* from the drug-susceptible *C. albicans* wt strain also abolishes *MDR1* expression in the presence of inducing chemicals, such as benomyl and hydrogen peroxide. This fact demonstrates that the transcription factor *MRR1* mediates both: an inducible *MDR1* expression in drug-susceptible strains and a constitutive *MDR1* overexpression in drug-resistant strains [26]. 

### 3.5. Expression of the efflux pump ABCB1 gene in HEK-293

Efflux pump ABCB1 protects mammalian organisms by extruding various xenobiotics and drugs from cells. Under pathological conditions, the overexpression of *ABCB1* in tumor cells results in multidrug resistance [28]. As determined by the measurements of Et^+^ fluorescence, CSDP^+^ at low concentrations alone or in combination with fluconazole did not show any considerable effect on the efflux activity in HEK-293 cells (Figure 3). Despite this, we checked the effect of CSDP^+^ on the expression of the *ABCB1* gene in HEK-293 cells. The results of experiments indicated that after the treatment of cells with 0.25 µg/mL of CSDP^+^, the expression of the *ABCB1* gene in HEK-293 cells was 1.99 ± 0.15-fold higher (Figure 5). Fluconazole at a concentration of 0.0625 µg/mL caused a 1.18 ± 0.05-fold increase, and the combination of both compounds at low concentrations resulted in a 1.11 ± 0.08-fold increase in the *ABCB1* expression (Figure 5).

In summary, the expression of the efflux pump gene *ABCB1* is only weakly increased after the exposure of HEK-293 cells to the tested compounds. Only a slight increase in *ABCB1* expression and low fluorescence of Et^+^ ions registered during the efflux measurements indicate very limited damage of the plasma membrane caused by CSDP^+^ and fluconazole at the concentrations used. 

## 4. Conclusions

Our experiments demonstrated that CSDP^+^ at low concentrations alone or in combination with fluconazole was not toxic to HEK-293 cells. Moreover, this compound decreased the ability of *C. albicans* to adhere to HEK-293 cells. Fluorescence analysis of efflux activity showed that mammalian cells were able to protect themselves from low concentrations of CSDP^+^ or fluconazole. The expression level of the efflux pump gene *ABCB1* in HEK-293 cells only slightly increased after the exposure of cells to the tested agents.

Treatment with CSDP^+^ in combination with fluconazole resulted in a considerable overexpression of the *MDR1* and *MRR1* genes in *C. albicans* wt and clinical isolate strains. These findings suggest that the expression of *MDR1* and *MRR1* could be associated with fluconazole resistance related efflux, and our next step would be the analysis of fungicidal mechanism of CSDP^+^ action in a wider range of clinical *C. albicans* and non-*albicans* isolates with a different degree of resistance.

## Figures and Tables

**Figure 1 microorganisms-09-00048-f001:**
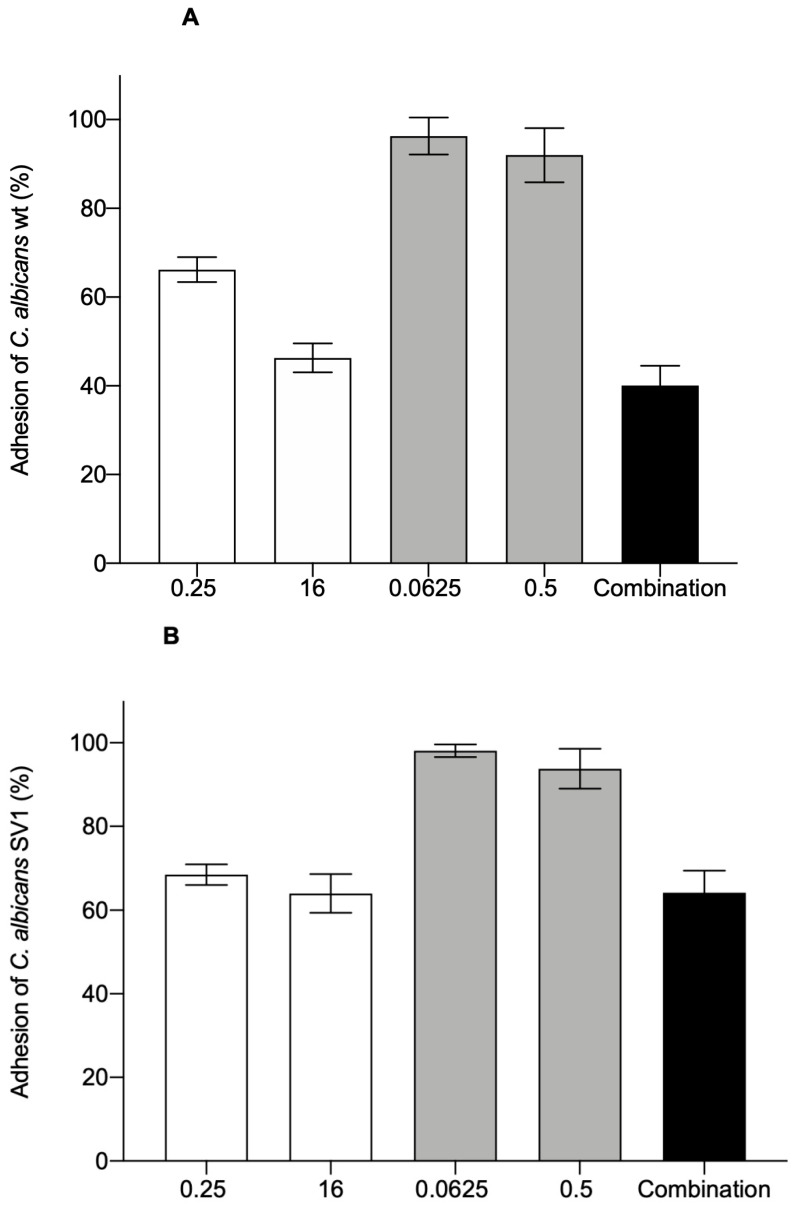
Adhesion of *C. albicans* wt (**A**) and SV1 (**B**) yeasts to the surface of HEK-293 cells. Cells were treated with different concentrations (µg/mL) of CSDP^+^ (white bars), fluconazole (grey bars), and their combination (0.25 µg/mL of CSDP^+^ and 0.0625 µg/mL of fluconazole, black bars). Data are shown as mean ± standard deviation (SD). In the absence of any compounds, 100% adhesion was registered.

**Figure 2 microorganisms-09-00048-f002:**
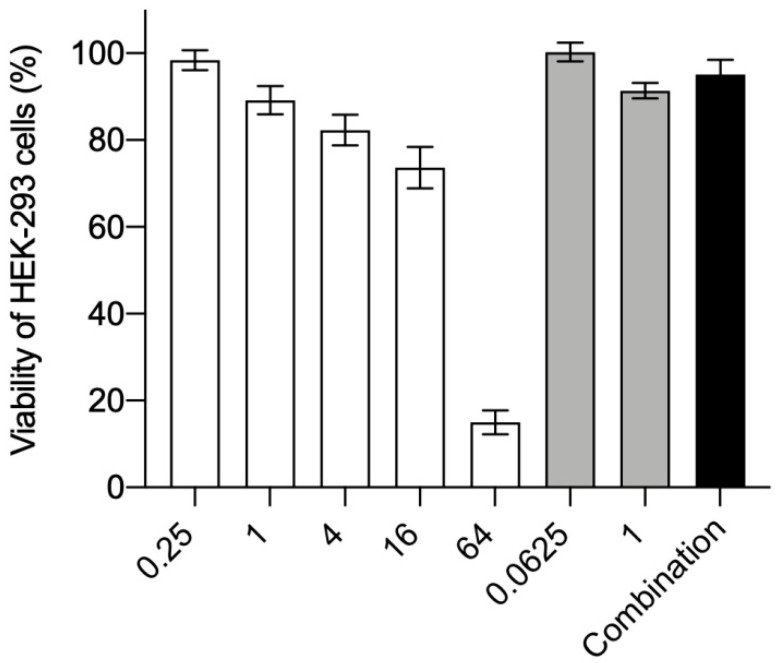
Effect of CSDP^+^ and fluconazole on the viability of HEK-293 cells. The cells were treated with CSDP+ (white bars), fluconazole (grey bars), and their combination (0.0625 µg/mL of fluconazole and 0.25 µg/mL of CSDP^+^, black bar). Data are shown as mean ± standard deviation (SD). The level of 100% corresponds to the viability of HEK-293 cells not treated with any compound.

**Figure 3 microorganisms-09-00048-f003:**
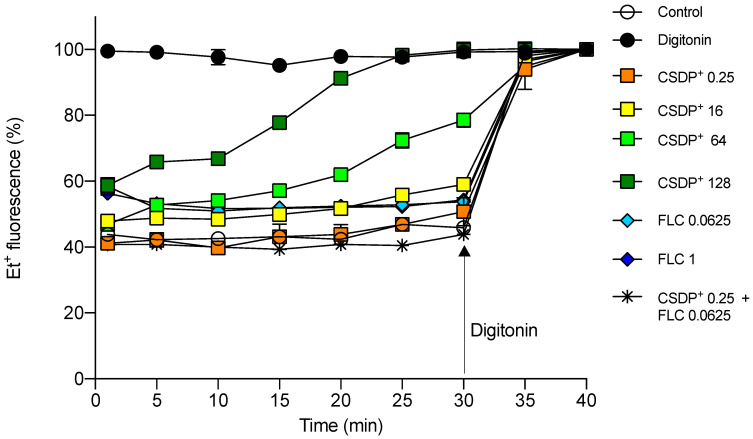
Et^+^ fluorescence in HEK-293 cell suspensions. Fluorescence of Et^+^ was registered in the presence of different concentrations (µg/mL) of CSDP^+^ and fluconazole (FLC). In control measurements, the cells were not treated with any compound or the medium contained 0.1 mg/mL of a lysing agent digitonin. Data are shown as mean ± standard deviation (SD).

**Figure 4 microorganisms-09-00048-f004:**
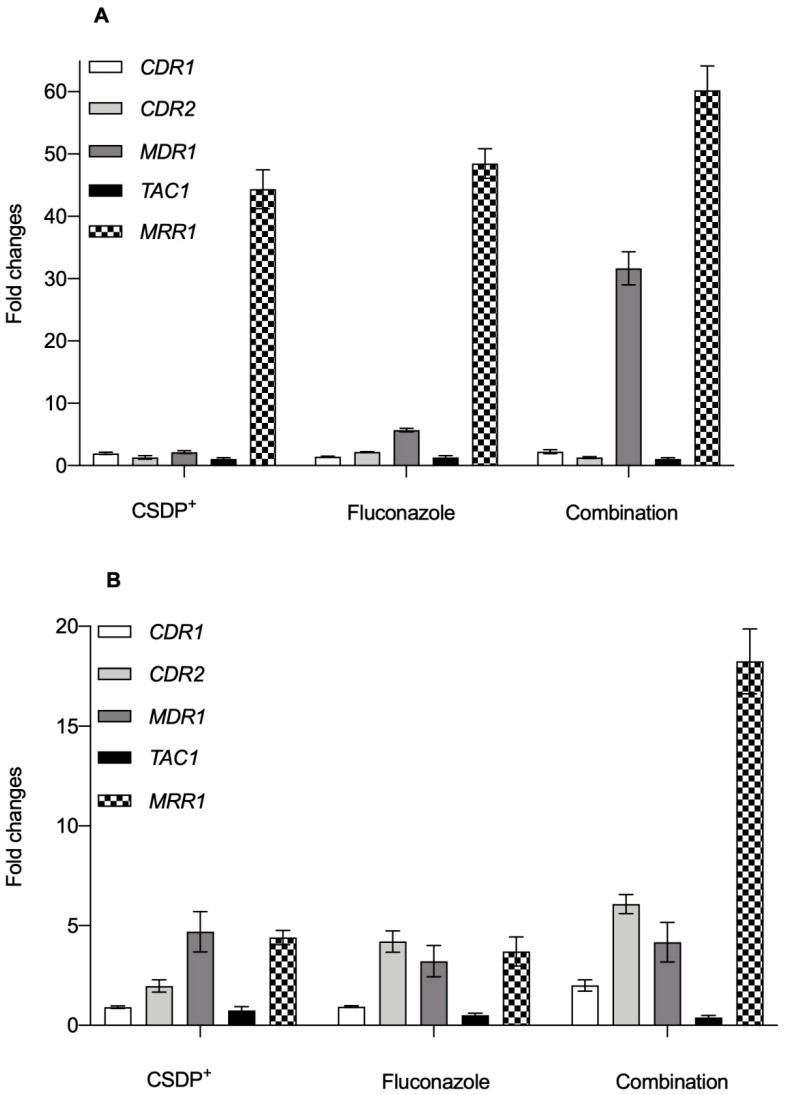
Gene expression in *C. albicans* wt (**A**) and SV1 (**B**) strain cells. The cells were treated with 2 µg/mL of CSDP^+^, 0.125 µg/mL of fluconazole or combination of both agents (0.25 µg/mL of CSDP^+^ and 0.0625 µg/mL of fluconazole). Data are shown as a mean fold change in gene expression relative to control nontreated *C. albicans* wt or SV1 cells.

**Figure 5 microorganisms-09-00048-f005:**
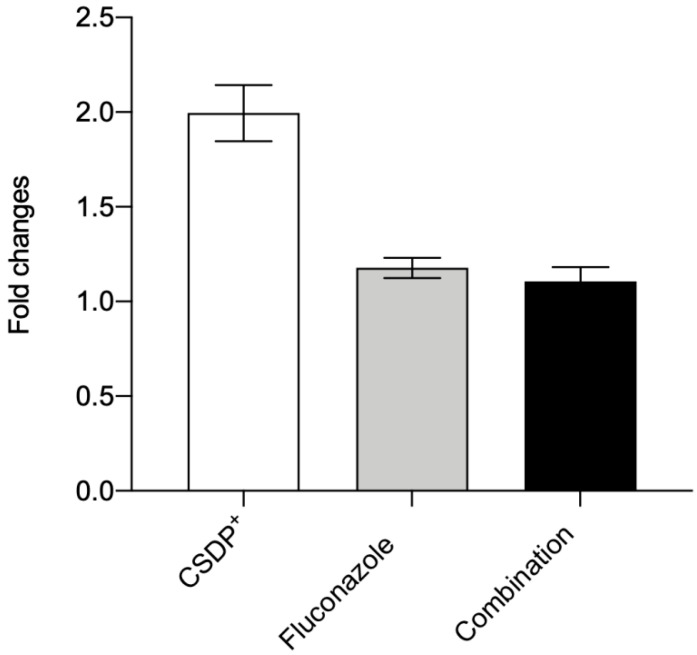
Expression of the *ABCB1* gene in HEK-293 cells. The cells were treated with CSDP^+^ (0.25 µg/mL), fluconazole (0.0625 µg/mL), or a combination of both agents (0.25 µg/mL of CSDP^+^ and 0.0625 µg/mL of fluconazole). Data are shown as a mean fold change in the gene expression relative to control nontreated HEK-293 cells.

**Table 1 microorganisms-09-00048-t001:** Primers used in the study.

*CDR1*	5′-GTACTATCCATCAACCATCAGCACTT-3′ (forward)
	5′-GCCGTTCTTCCACCTTTTTGTA-3′ (reverse)
*CDR2*	5′-TGCTGAACCGACAGACTCAGTT-3′ (forward)
	5′-AAGAGATTGCCAATTGTCCCATA-3′ (reverse)
*MDR1*	5′-TCAGTCCGATGTCAGAAAATGC-3′ (forward)
	5′-GCAGTGGGAATTTGTAGTATGACAA-3′ (reverse)
*TAC1*	5′-GAAATTGTTAATGACGGTTCTACCTTC-3′ (forward)
	5′-TATTCATATACCCAACCGGAAATTGG-3′ (reverse)
*MRR1*	5′-AACGCTGGTTATGGGTGA-3′ (forward)
	5′-TTTGCTGTTGGGCTTCTT-3′ (reverse)
*18S*	5′-GGATTTACTGAAGACTAACTACTG-3′ (forward)
	5′-GAACAACAACCGATCCCTAGT-3′ (reverse)
*GAPDH*	5′-GTCTCCTCTGACTTCAACAGCG-3′ (forward)
	5′-ACCACCCTGTTGCTGTAGCCAA-3′ (reverse)
*ABCB1*	5′-GTCCCAGGAGCCCATCCT-3′ (forward)
	5′-CCCGGCTGTTGTCTCCATA-3′ (reverse)

## Data Availability

The data presented in this study are available on request from the corresponding author.

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
