# Peer review of "Interaction of Styrylpyridinium Compound with Pathogenic Candida albicans Yeasts and Human Embryonic Kidney HEK-293 Cells"

_microorganisms, 2020, doi:10.3390/microorganisms9010048_

Round 1

Reviewer 1 Report

The paper from Simona Vaitkienėa et al describes the effect of newly synthesezed CSDP+ with C.albicans and HEK cells being non toxic for the latter while reducing the adhesion of fungi. Taking onto account wide-spread biofilm-associated candidiasis the obtained results are defenitely of interest. Nevertheless, some points should be clarified before paper consideration for publication.

This is not clear what is the link between adhesion/biofilm formation and efflux pumps? If cells growth is inhibited, generally no biofilm can be developed.

In the end of introduction, there would be fine to include a sentence with conclusion describing the main message of the paper.

This is not clear why HEK-293 cells were used here. Since most C.albicans biofilms are located on various mucosa and skin, human fibroblasts, keratinocytes of ephytelium cells would be more relevant.

Fig 1: I would say there is no effect of fluconazole on adhesion of clinical isolate, while the synergism is observed for the wt. Whta is the difference in fluconazole and CSDP+ sensitivity of C.albicans wt and SV1 strains? It should be added in materials section and here too. As well, since there is a difference between strains, a series of clinical isolates should be tested in this experiment to show whether the proposed approach could be helpful in practice.

Why these concentrations of compounds (especially of fluconasole) have been used? What is the MIC and BPC (biofilm-preventing concentration)? Generally the checker-board assay should be performed. If these data were published, their description (FICI, the best ration of concentrations) should be included also here.

Fig 2: the same qustion as above: why these concentrations of compounds (especially of fluconasole) have been used? Authors should calculate CC50 values for each compound and ther combination (perhaps, leading to the best outcome). Look here https://www.researchgate.net/post/How-to-calculate-CC50-at-graphpad-prism

Fig 3: Please explain the aim of Digitonin addition and observed effect.

Fig 4: Why the induction is so different if two strains? I suppose this is becase of different susceptibility to the drugs. Therefore the concentrations of drugs should be compared to MIC, and their ratio should be the same for both strains. Also this is unclear why 8-fold less CSDP+ is present in combination with FLC in compare to solely CSDP+.

As well the Y-axys could be in log-scale (log2) and should be of same values on both figures (a and b)

Finally, on all figures SD should be shown instead od SEM.

Author Response

Reviewer: 1

Author response

This is not clear what is the link between adhesion/biofilm formation and efflux pumps? If cells growth is inhibited, generally no biofilm can be developed.

Mukherjee et al. showed that efflux pumps play a critical role in azole resistance in early-phase of biofilms formation DOI: 10.1128/iai.71.8.4333-4340.2003. They determined the antifungal susceptibilities of biofilms formed by mutants carrying single, double, or triple deletion mutations of the CDR and MDR1 genes. Their results showed that at the early phase of development, biofilms formed by these mutants were more susceptible to fluconazole than those formed by the wild-type strain.

In the end of introduction, there would be fine to include a sentence with conclusion describing the main message of the paper.

At the end of introduction, we included a sentence summarizing the results.

This is not clear why HEK-293 cells were used here. Since most C.albicans biofilms are located on various mucosa and skin, human fibroblasts, keratinocytes of ephytelium cells would be more relevant.

Pathogenic Candida yeasts not only affect the skin, but also cause serious systemic infections. Human kidney cells (HEK-293) are successfully used as a model of mammalian cells in such studies (DOI: 10.1007/s11696-017-0141-8; doi: 10.1128/AAC.00317-16).

Fig 1: I would say there is no effect of fluconazole on adhesion of clinical isolate, while the synergism is observed for the wt. Whta is the difference in fluconazole and CSDP+ sensitivity of C.albicans wt and SV1 strains? It should be added in materials section and here too. As well, since there is a difference between strains, a series of clinical isolates should be tested in this experiment to show whether the proposed approach could be helpful in practice.

Susceptibility of WT and SV1 cells to fluconazole and styrylpyridinium compounds was analyzed in our previous study, which is cited in the text doi: 10.1111/cbdd.13777.

In our future experiments, we are going to focus on the fungicidal mechanism of CSDP+ action and test it in a wider range of clinical C. albicans and non-albicansisolates with a different degree of resistance.

Why these concentrations of compounds (especially of fluconasole) have been used? What is the MIC and BPC (biofilm-preventing concentration)? Generally the checker-board assay should be performed. If these data were published, their description (FICI, the best ration of concentrations) should be included also here.

All the concentrations used in this study were selected according to results of our previous study. A sentence about this was added to the part 3. Results and Discussion.

Fig 2: the same qustion as above: why these concentrations of compounds (especially of fluconasole) have been used? Authors should calculate CC50 values for each compound and ther combination (perhaps, leading to the best outcome). Look here https://www.researchgate.net/post/How-to-calculate-CC50-at-graphpad-prism

The same answer as above.

Fig 3: Please explain the aim of Digitonin addition and observed effect.

Digitonin is a detergent permeabilizing the plasma membrane. The maximum binding of Et+ to DNA is achieved after digitonin addition, leading to the maximum fluorescence of this lipophilic ion.

Fig 4: Why the induction is so different if two strains? I suppose this is becase of different susceptibility to the drugs. Therefore the concentrations of drugs should be compared to MIC, and their ratio should be the same for both strains. Also this is unclear why 8-fold less CSDP+ is present in combination with FLC in compare to solely CSDP+.

According to our previous study, the minimal concentrations of CSDP+ and FLC effective against the studied Candida cells were selected and used in the experiments. 

As well the Y-axis could be in log-scale (log2) and should be of same values on both figures (a and b)

We are very thankful for your suggestion. In our opinion, the data shown in these figures are clear and easy to understand, to our mind log scale would complicate the presentation of results. Therefore, we left the figures as they were.

Finally, on all figures SD should be shown instead od SEM.

After the revision, in all the figures, the measure of dispersion is presented as SD instead of SEM. The text was amended as well.

Reviewer 2 Report

The Authors investigated the interaction of styrylpyridinium compound with pathogenic Candida albicans yeasts and human embryonic kidney HEK-293 cells. They tested the effect of styrylpyridinium alone and in combination (checkerboard method) with fluconazole. The work is interesting and well written. Besides, it provides a valid contribution to the problem of resistance of Candida albicans to commonly used antifungal agents. In my opinion, in this current version, the manuscript is suitable for publication.

Author Response

We would like to thank the Reviewer for the comments and for classification of the manuscript as suitable for publication.

Reviewer 3 Report

This manuscript is describing about interaction of styrylpyridinium compound with pathogenic Candida albicans and human embryonic kidney cells. The object was clear and the manuscript was wriiten well. However, English can be improved for better expression and several minor points should be revised before final decision.

  • Abs, Should be written in one paragraph.
  • L14, alone and -> alone or
  • L23, The result of 'treatment with CSDP+ alone' should be mentioned. ex, ... genes, though that with CSDP+ alone showed xxxxx..
  • Italic forms of genes in the text should be chcked carefully.
  • Font sizes of figures were different hugely, and unified in font sizes in labels and titles of the figures.
  • In Figure 3, two symbols were wrong. It appears as squares, not ug.

Author Response

Reviewer: 3

Abs, Should be written in one paragraph.

The abstract is corrected to one paragraph following the reviewer’s suggestion.

L14, alone and -> alone or

We corrected the text following the reviewer’s suggestion.

L23, The result of 'treatment with CSDP+ alone' should be mentioned. ex, ... genes, though that with CSDP+ alone showed xxxxx..

In the abstract we focused on the most important finding -   that C. albicans treated with CSDP+ in combination with fluconazole resulted in a considerable overexpression of the MDR1 and MRR1 genes.

Italic forms of genes in the text should be checked carefully.

In lines 50 and 51, TAC1 is mentioned as a transcription factor, not as a gene. Therefore, it is not written in Italic.

Font sizes of figures were different hugely, and unified in font sizes in labels and titles of the figures.

The text font size in all figures was corrected according to the reviewer’s suggestion.

In Figure 3, two symbols were wrong. It appears as squares, not ug.

Figure 3 was amended to make it clearer.

Round 2

Reviewer 1 Report

Authors answered all my previous questions, nevertheless, to get a readble paper, I suggest that
- Despite the susceptibility of WT and SV1 cells to fluconazole and styrylpyridinium compounds was analyzed in previous study, these data must be shown here - MIC, BPC, and synergism testing results. Readers dont wont to search previous paper only for several values. This is strictly required for correct interpretation of Fig 1 and Fig 4.
- Please mention tne CC50 for compounds of ineterest, this is important to know their toxicity.
- the described effect of digitonin shou;d be included into the paper, not only in the answer.

Author Response

Reviewer: 1

Author response

Authors answered all my previous questions, nevertheless, to get a readble paper, I suggest that 
- Despite the susceptibility of WT and SV1 cells to fluconazole and styrylpyridinium compounds was analyzed in previous study, these data must be shown here - MIC, BPC, and synergism testing results. Readers dont wont to search previous paper only for several values. This is strictly required for correct interpretation of Fig 1 and Fig 4.
- Please mention tne CC50 for compounds of ineterest, this is important to know their toxicity.
- the described effect of digitonin shou;d be included into the paper, not only in the answer.

Several sentences describing data of our previous study are added to the part 3. Results and Discussion.

Explanation of digitonin effect is added to the part 3.3. Effect of CSDP+ and fluconazole on Et+ fluorescence in HEK-293 cell suspensions.